# A Near-Optimal Algorithm for Debiasing Trained Machine Learning Models

## Abstract

We present an efficient and scalable algorithm for debiasing trained models, including deep neural networks (DNNs), which we prove to be near-optimal by bounding its excess Bayes risk. Unlike previous black-box reduction methods to cost-sensitive classification rules, the proposed algorithm operates on models that have been trained without having to retrain the model. Furthermore, as the algorithm is based on projected stochastic gradient descent (SGD), it is particularly attractive for deep learning applications. We empirically validate the proposed algorithm on standard benchmark datasets across both classical algorithms and modern DNN architectures and demonstrate that it outperforms previous post-processing approaches for unbiased classification.

## 1 Introduction

Machine learning is increasingly applied to critical decisions which can have a lasting impact on individual lives, such as for credit lending (Bruckner, 2018), medical applications (Deo, 2015), and criminal justice (Brennan et al., 2009). Consequently, it is imperative to understand and improve the degree of bias of such automated decision-making.

Unfortunately, despite the fact that bias (or "fairness") is a central concept in our society today, it is difficult to define it in precise terms. In fact, as people perceive ethical matters differently depending on a plethora of factors including geographical location or culture (Awad et al., 2018), no universally-agreed upon definition for bias exists. Moreover, the definition of bias may depend on the application and might even be ignored in favor of accuracy when the stakes are high, such as in medical diagnosis (Kleinberg et al., 2017; Ingold and Soper, 2016). As such, it is not surprising that several definitions of "unbiased classification" have been introduced. These include statistical parity (Dwork et al., 2012; Zafar et al., 2017a), equality of opportunity (Hardt et al., 2016), and equalized odds (Hardt et al., 2016; Kleinberg et al., 2017). Unfortunately, such definitions are not generally compatible (Chouldechova, 2017) and some might even be in conflict with calibration (Kleinberg et al., 2017). In addition, because fairness is a societal concept, it does not necessarily translate into a statistical criteria (Chouldechova, 2017; Dixon et al., 2018).

**Statistical parity** Let $\mathcal{X}$ be an instance space and let $\mathcal{Y} = \{0, 1\}$ be the target set in a standard binary classification problem. In the fair classification setting, we may further assume the existence of a (possibly randomized) sensitive attribute $s : \mathcal{X} \rightarrow \{0, 1, \ldots, K\}$, where $s(x) = k$ if and only if $x \in X_k$ for some total partition $\mathcal{X} = \cup_k X_k$. For example, $\mathcal{X}$ might correspond to the set of job applicants while $s$ indicates their gender. Here, the sensitive attribute can be randomized if, for instance, the gender of an applicant is not a deterministic function of the full instance $x \in \mathcal{X}$ (e.g. number of publications, years of experience, ...etc). Then, a commonly used criterion for fairness is to require similar mean outcomes across the sensitive attribute. This property is well-captured through the notion of statistical parity (a.k.a. demographic parity) (Corbett-Davies et al., 2017; Dwork et al., 2012; Zafar et al., 2017a; Mehrabi et al., 2019):

**Definition 1** (Statistical Parity). *Let $\mathcal{X}$ be an instance space and $\mathcal{X} = \cup_k X_k$ be a total partition of $\mathcal{X}$. A classifier $f : \mathcal{X} \rightarrow \{0, 1\}$ satisfies $\epsilon$ statistical parity across all groups $X_1, \ldots, X_K$ if:*

$$\max_{k \in \{1,2,\ldots,K\}} \mathbb{E}_{\boldsymbol{x}}[f(\boldsymbol{x}) \,|\, \boldsymbol{x} \in X_k] \; - \; \min_{k \in \{1,2,\ldots,K\}} \mathbb{E}_{\boldsymbol{x}}[f(\boldsymbol{x}) \,|\, \boldsymbol{x} \in X_k] \leq \epsilon$$

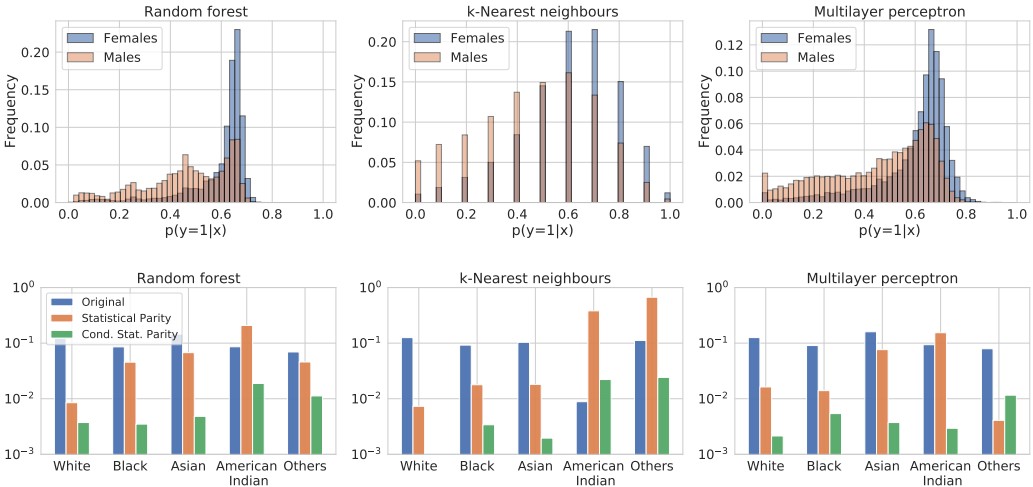

Figure 1: **Top:** Histogram of classifiers' predictions on both subpopulations, demonstrating a clear gender bias in all cases. **Bottom:** The bias defined as the absolute difference in mean outcome between genders within different demographic groups, before and after applying the proposed algorithm. Blue bars show the results of the unmodified classifier, orange bars show the results of optimizing for statistical parity with no regard for demographic information. Finally, green bars are the results of applying statistical parity on the cross product of gender and ethnicity.

To motivate and further clarify the definition, we showcase the empirical results on the Adult benchmark dataset (Blake and Merz, 1998) in Figure 1. When tasked with predicting whether the income of individuals is above $50K per year, all considered classifiers exhibit gender-related bias. One way of removing such bias is to enforce statistical parity across genders. Crucially, however, without taking ethnicity into account, different demographic groups may experience different outcomes. In fact, gender bias can actually *increase* in some minority groups after enforcing statistical parity. This can be fixed by redefining the sensitive attribute to be the *cross product* of both gender and ethnicity (green bars).

Our main contribution is to present a near-optimal recipe for debiasing models, including deep neural networks, according to Definition 1. Specifically, we formulate the task of debiasing learned models as a regularized optimization problem that is solved efficiently using the projected SGD method. We show how the algorithm produces thresholding rules with randomization near the thresholds, where the width of randomization is controlled by the regularization parameter. We also show that randomization near the threshold is necessary for Bayes risk consistency. While we focus on binary sensitive attributes in our experiments in Section 5, our algorithm and its theoretical guarantees continue to hold for non-binary sensitive attributes as well.

**Statement of Contribution**

1. We derive a near-optimal post-processing algorithm for debiasing learned models (Section 3).
2. We prove theoretical guarantees for the proposed algorithm, including a proof of correctness and an explicit bound on the Bayes excess risk (Section 4).
3. We empirically validate the proposed algorithm on benchmark datasets across both classical algorithms and modern DNN architectures. Our experiments demonstrate that the proposed algorithm significantly outperforms previous post-processing methods (Section 5).

In Appendix E, we also show how the proposed algorithm can be modified to handle other criteria of bias as well.

## 2 RELATED WORK

Algorithms for fair machine learning can be broadly classified into three groups: (1) pre-processing methods, (2) in-processing methods, and (3) post-processing methods (Zafar et al., 2019).

Preprocessing algorithms transform the data into a different representation such that any classifier trained on it will not exhibit bias. This includes methods for learning a fair representation (Zemel et al., 2013; Lum and Johndrow, 2016; Bolukbasi et al., 2016; Calmon et al., 2017; Madras et al., 2018; Kamiran and Calders, 2012), label manipulation (Kamiran and Calders, 2009), data augmentation (Dixon et al., 2018), or disentanglement (Locatello et al., 2019).

On the other hand, in-processing methods constrain the behavior of learning algorithms in order to control bias. This includes methods based on adversarial learning (Zhang et al., 2018) and constraint-based classification, such as by incorporating constrains on the decision margin (Zafar et al., 2019) or features (Grgić-Hlača et al., 2018). Agarwal et al. (2018) showed that the task of learning an unbiased classifier could be reduced to a *sequence* of cost-sensitive classification problems, which could be applied to any *black-box* classifier. One caveat of the latter approach is that it requires solving a linear program (LP) and retraining classifiers, such as neural networks, *many* times before convergence.

The algorithm we propose in this paper is a post-processing method, which can be justified theoretically (Corbett-Davies et al., 2017; Hardt et al., 2016; Menon and Williamson, 2018; Celis et al., 2019). Fish et al. (2016) and Woodworth et al. (2017) fall under this category. However, the former only provides generalization guarantees without consistency results while the latter proposes a two-stage approach that requires changes to the original training algorithm. Kamiran et al. (2012) also proposes a post-processing algorithm, called Reject Option Classifier (ROC), without providing any theoretical guarantees. In contrast, our algorithm is Bayes consistent and does not alter the original classification method. In Celis et al. (2019) and Menon and Williamson (2018), instance-dependent thresholding rules are also learned. However, our algorithm also learns to *randomize* around the threshold (Figure 2(a)) and this randomization is *key* to our algorithm both theoretically as well as experimentally (Appendix C and Section 5). Hardt et al. (2016) learns a randomized post-processing rule but our proposed algorithm outperforms it in all of our experiments (Section 5).

Woodworth et al. (2017) showed that the post-processing approach can, sometimes, be highly suboptimal. Nevertheless, the latter result does not contradict the statement that our post-processing rule is near-optimal because we assume that the original classifier outputs a monotone transformation of some approximation to the posterior probability $p(\mathbf{y} = 1 \mid \mathbf{x})$ (e.g. margin or softmax output) whereas Woodworth et al. (2017) assumed in their construction that the post-processing rule had access to the binary predictions only.

We argue that the proposed algorithm has distinct advantages, particularly for deep neural networks (DNNs). First, stochastic convex optimization methods are well-understood and can scale well to massive amounts of data (Bottou, 2010), which is often the case in deep learning today. Second, the guarantees provided by our algorithm hold w.r.t. the *binary* predictions instead of using a proxy, such as the margin as in some previous works (Zafar et al., 2017b; 2019). Third, unlike previous reduction methods that would require retraining a deep neural network several times until convergence (Agarwal et al., 2018), which can be prohibitively expensive, our algorithm operates on learned models that are trained once and does not require retraining.

Besides developing algorithms for fair classification, several recent works focused on other related aspects, such as proposing new definitions for fairness; e.g. demographic parity (Dwork et al., 2012; Mehrabi et al., 2019), equalized odds (Hardt et al., 2016), equality of opportunity/disparate mistreatment (Zafar et al., 2017a; Hardt et al., 2016), and individual fairness (Dwork et al., 2012). Recent works have also established several impossibility results related to fair classification, such as Kleinberg et al. (2017); Chouldechova (2017). In our case, we derive a new impossibility result that holds for *any* deterministic binary classifier and relate it to the task of controlling the covariance between the classifier's predictions and the sensitive attribute (Appendix E).

## 3   Near-optimal algorithm for Statistical Parity

**Notation**   We reserve boldface letters for random variables (e.g. $\mathbf{x}$), small letters for instances (e.g. $x$), capital letters for sets (e.g. $X$), and calligraphic typeface for universal sets (e.g. the instance space $\mathcal{X}$). Given a set $S$, $1_S(x) \in \{0, 1\}$ is the characteristic function indicating whether $x \in S$. We denote by $[n]$ the set of integers $\{1, \ldots, n\}$ and $[x]^+ = \max\{0, x\}$.

**Algorithm**    Given a classifier $f : \mathcal{X} \to [-1, +1]$ our goal is to post-process the predictions made by $f$[1] in order to control the bias with respect to a sensitive attribute $s : \mathcal{X} \to [K]$ as in Definition 1. To this end, instead of learning a deterministic classifier, we consider *randomized* prediction rules of the form

$$\tilde{h} : \mathcal{X} \times \{1, 2, \ldots, K\} \times [-1, 1] \to [0, 1],$$

where $\tilde{h}(\mathbf{x})$ represents the probability of predicting the positive class given (i) instance $\mathbf{x} \in \mathcal{X}$, (ii) sensitive attribute $s(\mathbf{x})$, and (iii) classifier's output $f(\mathbf{x})$.

As discussed in Appendix B, for post-processing rule $\tilde{h}(x)$, and for each group $X_k \subseteq \mathcal{X}$, the fairness constraint in Definition 1 can be written as $|\mathbb{E}_{\mathbf{x}}[\tilde{h}(\mathbf{x}) \mid \mathbf{x} \in X_k] - \rho| \leq \epsilon$, where $\rho \in [0, 1]$ is a hyper-parameter tuned via a validation dataset. On the other hand, minimizing the probability of altering the predictions of the original classifier can be achieved by maximizing the inner product $\mathbb{E}_{\mathbf{x}}[\tilde{h}(\mathbf{x}) \cdot f(\mathbf{x})]$. Instead of optimizing this quantity directly, which would lead to a pure thresholding rule, we minimize the *regularized* objective: $(\gamma/2)\mathbb{E}_{\mathbf{x}}[\tilde{h}(\mathbf{x})^2] - \mathbb{E}_{\mathbf{x}}[\tilde{h}(\mathbf{x}) \cdot f(\mathbf{x})]$ for some regularization parameter $\gamma > 0$. This regularization leads to randomization around the threshold, which we show to be critical, both theoretically (Section 4 and Appendix C) and experimentally (Section 5). Using Lagrange duality we show that the solution reduces to the update rules in Equation 2 with optimization variables $\{\lambda_k, \mu_k\}_{k \in [K]}$ and the corresponding predictor which outputs $+1$ for group $X_k$ with probability $\tilde{h}_\gamma(x)$ is given by

$$\tilde{h}_\gamma(x) = \begin{cases} 0, & f(x) \leq \lambda_k - \mu_k \\ (f(x) - \lambda_k + \mu_k)/\gamma, & \lambda_k - \mu_k \leq f(x) \leq \lambda_k - \mu_k + \gamma \\ 1, & f(x) \geq \lambda_k - \mu_k + \gamma \end{cases} \tag{1}$$

where $\xi_\gamma$ is given by Eq. (3).

**Update rules**    To learn these parameters, one can apply the following update rules (Appendix B):

$$\lambda_{s(\mathbf{x})} \leftarrow \max \left\{ 0, \ \lambda_{s(\mathbf{x})} - \eta \left( \frac{\epsilon}{2} + \rho + \frac{\partial}{\partial \lambda_{s(\mathbf{x})}} \xi_\gamma \big( f(\mathbf{x}) - (\lambda_{s(\mathbf{x})} - \mu_{s(\mathbf{x})}) \big) \right) \right\}$$

$$\mu_{s(\mathbf{x})} \leftarrow \max \left\{ 0, \ \mu_{s(\mathbf{x})} - \eta \left( \frac{\epsilon}{2} - \rho + \frac{\partial}{\partial \mu_{s(\mathbf{x})}} \xi_\gamma \big( f(\mathbf{x}) - (\lambda_{s(\mathbf{x})} - \mu_{s(\mathbf{x})}) \big) \right) \right\}, \tag{2}$$

where, again, $\rho \in [0, 1]$ is a hyperparameter tuned via a validation dataset, $s : \mathcal{X} \to [K]$ is the sensitive attribute, and $\gamma > 0$ is a regularization parameter that controls the level of randomization. In addition, the function $\xi_\gamma : \mathbb{R} \to \mathbb{R}^+$ is given by:

$$\xi_\gamma(w) = \frac{w^2}{2\gamma} \cdot \mathbb{I}\{0 \leq w \leq \gamma\} + \left( w - \frac{\gamma}{2} \right) \cdot \mathbb{I}\{w > \gamma\} \tag{3}$$

Note that $\xi_\gamma$ is convex and its derivative $\xi_\gamma'$ is $(1/\gamma)$-Lipschitz continuous; it can be interpreted as differentiable approximation to the ReLU unit (Nair and Hinton, 2010). A full pseudocode of the proposed algorithm is presented in Appendix A.

## 4    THEORETICAL ANALYSIS

Next, we analyze the algorithm. Our first theoretical result is to show that the prediction rule in Equation 1 learned through the update rules presented in Section 3 satisfies the desired fairness guarantees on the training sample.

**Theorem 1** (Correctness). *Let $\tilde{h}_\gamma : \mathcal{X} \to [0, 1]$ be the randomized predictor in Equation 1 learned by applying the update rules in Equation 2 starting with $\mu_k = 0, \lambda_k = 0, \forall k \in [K]$ until convergence. Then, $\tilde{h}_\gamma$ satisfies $\epsilon$ statistical parity w.r.t. $\{X_k\}_{k \in [K]}$ in the training sample.*

The proof of Theorem 1 is presented in Appendix B. The following guarantee, which holds w.r.t. the underlying data distribution, shows that the randomized prediction rule converges to the Bayes

---

[1]Ideally an estimate of some monotone transformation of $2\eta(x) - 1$, where $\eta(x) = p(\mathbf{y} = 1|\mathbf{x} = x)$ is the Bayes regressor. This is not a strong assumption because many algorithms can be calibrated to provide probability scores (Platt et al., 1999; Guo et al., 2017).

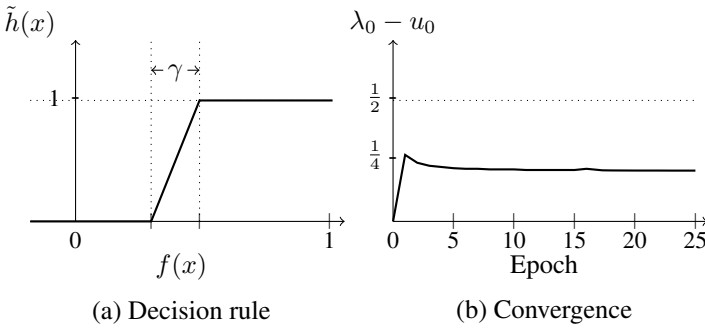

(a) Decision rule            (b) Convergence

Figure 2: (a) The learned post-processing decision rule $\tilde{h}(x)$ in Equation 1 as a function of the classifier's score $f(x)$. Randomization is applied when $\tilde{h}(x) \in (0,1)$, which can be controlled using the regularization parameter $\gamma > 0$. (b) The value of $\lambda_0 - u_0$ is plotted against the number of epochs in the projected SGD method applied to the output of the random forests classifier trained on the Adult dataset to implement statistical parity with respect to the gender attribute (cf. Section 5 and Figure 1). We observe fast convergence in agreement with Proposition 1.

optimal unbiased classifier if the original classifier $f$ is Bayes consistent. The proof of the following theorem (Appendix C) is based on the Lipschitz continuity of the decision rule when $\gamma > 0$ and the robustness framework of Xu and Mannor (2010).

**Theorem 2.** *Let $h^\star = \arg\min_{h \in \mathcal{H}_\epsilon} \mathbb{E}[h(\boldsymbol{x}) \neq \boldsymbol{y}]$, where $\mathcal{H}_\epsilon$ is the set of binary predictors on $\mathcal{X}$ that satisfy fairness according to Definition 1 for $\epsilon > 0$. Let $\tilde{h}_\gamma : \mathcal{X} \to [0,1]$ be the randomized learning rule in Equation 1. If $\tilde{h}_\gamma$ is trained on a freshly sampled data of size $N$, then there exists a value of $\rho \in [0,1]$ such that the following holds with a probability of at least $1 - \delta$:*

$$\mathbb{E}[\mathbb{I}\{\tilde{h}_\gamma(\boldsymbol{x}) \neq \boldsymbol{y}\}] \leq \mathbb{E}[\mathbb{I}\{h^\star(\boldsymbol{x}) \neq \boldsymbol{y}\}] + \mathbb{E}\left|2\eta(\boldsymbol{x}) - 1 - f(\boldsymbol{x})\right| + 2\gamma + \frac{8(2 + \frac{1}{\gamma})}{N^{\frac{1}{3}}} + 4\sqrt{\frac{2K + 2\log\frac{2}{\delta}}{N}},$$

*where $\eta(x) = p(\boldsymbol{y} = 1 | \boldsymbol{x} = x)$ is the Bayes regressor and $K$ is the number of groups $X_k$.*

Consequently, if the original classifier is Bayes consistent and we have: $N \to \infty$, $\gamma \to 0^+$ and $\gamma N^{-\frac{1}{3}} \to \infty$, then $\mathbb{E}[\tilde{h}_\gamma(\mathbf{x}) \neq \boldsymbol{y}] \xrightarrow{P} \mathbb{E}[h^\star(\mathbf{x}) \neq \boldsymbol{y}]$. Hence, the updates converge to the *optimal* prediction rule subject to the chosen fairness constraint.

**Running time** As shown in Appendix B, the update rules in Equation 2 perform a projected stochastic gradient descent on the following optimization problem:

$$\min_{(\mu_1, \lambda_1), \dots, (\mu_K, \lambda_K) \geq 0} F = \mathbb{E}_{\mathbf{x}}\left[\epsilon\left(\lambda_{s(\mathbf{x})} + \mu_{s(\mathbf{x})}\right) + \rho\left(\lambda_{s(\mathbf{x})} - \mu_{s(\mathbf{x})}\right) + \xi_\gamma(f(\mathbf{x}) - (\lambda_{s(\mathbf{x})} - \mu_{s(\mathbf{x})}))\right] \quad (4)$$

We assume with no loss of generality that $f(x) \in [-1, 1]$ since $f(x)$ is assumed to be an estimator to $2\eta(x) - 1$ (see Section 3 and Appendix B) and any thresholding rule over $f(x)$ can be transformed into an equivalent rule over a monotone increasing function of $f$ (i.e. using the hyperbolic tangent).

**Proposition 1.** *Let $\mu^{(0)} = \lambda^{(0)} = 0$ and write $\mu^{(t)}, \lambda^{(t)} \in \mathbb{R}^K$ for the value of the optimization variables after $t$ updates defined in Equation 2 for some fixed learning rate $\alpha_t = \alpha$. Let $\bar{\mu} = (1/T) \sum_{t=1}^{T} \mu^{(t)}(x)$ and $\bar{\lambda} = (1/T) \sum_{t=1}^{T} \lambda^{(t)}(x)$. Then,*

$$\mathbb{E}[\bar{F}] - F(\mu^\star)) \leq \frac{(1 + \rho + \epsilon)^2 \alpha}{2} + \frac{||\mu^\star||_2^2 + ||\gamma^\star||_2^2}{2T\alpha}, \quad (5)$$

*where $\bar{F} : \mathbb{R}^K \times \mathbb{R}^K \to \mathbb{R}$ is the objective function in (4) using the averaged solution $\bar{\mu}$ and $\bar{\lambda}$ while $F^\star$ is its optimal value. In particular, $\mathbb{E}[\bar{F}] - F(\mu^\star)) = \mathcal{O}(\sqrt{K/T})$ when $\alpha = \mathcal{O}(\sqrt{K/T})$.*

The proof is in Appendix D. Hence, the post-processing rule can be efficiently computed. In practice, we observe fast convergence as shown in Figure 2(b).

As shown in Figure 2(a), the hyperparameter $\gamma$ controls the width of randomization around the thresholds. A large value of $\gamma$ may reduce the accuracy of the classifier. On the other hand, $\gamma$

| | | Bias | | | |
|---|---|---|---|---|---|
| Dataset | Classifier | Original | Proposed | Hardt et al. (2016) | Shift Inference | ROC |
| ADULT | RF | $.38 \pm .02$ | $.01 \pm .003$ | $.01 \pm .003$ | $.16 \pm .011$ | $.02 \pm .002$ |
| | $k$NN | $.24 \pm .02$ | $.02 \pm .005$ | $.01 \pm .004$ | $.08 \pm .006$ | $.08 \pm .006$ |
| | MLP | $.29 \pm .02$ | $.01 \pm .002$ | $.02 \pm .002$ | $.10 \pm .012$ | $.02 \pm .003$ |
| | LR | $.39 \pm .02$ | $.01 \pm .004$ | $.02 \pm .007$ | $.10 \pm .020$ | $.01 \pm .002$ |
| CDDD | RF | $.07 \pm .03$ | $.01 \pm .002$ | $.01 \pm .002$ | $.09 \pm .010$ | $.02 \pm .004$ |
| | $k$NN | $.10 \pm .02$ | $.01 \pm .005$ | $.01 \pm .001$ | $.18 \pm .005$ | $.02 \pm .005$ |
| | MLP | $.13 \pm .02$ | $.01 \pm .005$ | $.01 \pm .003$ | $.12 \pm .005$ | $.02 \pm .003$ |
| | LR | $.12 \pm .02$ | $.01 \pm .003$ | $.01 \pm .001$ | $.13 \pm .003$ | $.01 \pm .003$ |

| | | Test error | | | |
|---|---|---|---|---|---|
| Dataset | Classifier | Original | Proposed | Hardt et al. (2016) | Shift Inference | ROC |
| ADULT | RF | $34.1\pm.2\%$ | $35.8\pm.1\%$ | $38.0\pm.1\%$ | $34.5\pm.1\%$ | $36.0\pm.3\%$ |
| | $k$NN | $39.9\pm.2\%$ | $39.7\pm.2\%$ | $42.0\pm.2\%$ | $39.2\pm.2\%$ | $39.5\pm.2\%$ |
| | MLP | $34.8\pm.1\%$ | $36.8\pm.1\%$ | $37.4\pm.2\%$ | $35.7\pm.2\%$ | $36.3\pm.1\%$ |
| | LR | $35.3\pm.1\%$ | $36.6\pm.2\%$ | $38.8\pm.1\%$ | $35.9\pm.1\%$ | $36.5\pm.2\%$ |
| CDDD | RF | $18.8\pm.2\%$ | $18.2\pm.1\%$ | $19.4\pm.2\%$ | $19.0\pm.1\%$ | $18.5\pm.2\%$ |
| | $k$NN | $20.4\pm.2\%$ | $19.6\pm.2\%$ | $21.3\pm.1\%$ | $22.7\pm.2\%$ | $19.6\pm.2\%$ |
| | MLP | $19.5\pm.1\%$ | $18.7\pm.2\%$ | $21.2\pm.2\%$ | $19.4\pm.1\%$ | $19.0\pm.2\%$ |
| | LR | $19.4\pm.2\%$ | $18.3\pm.1\%$ | $21.7\pm.1\%$ | $19.8\pm.2\%$ | $19.5\pm.2\%$ |

Table 1: A comparison of the four post-processing methods on two datasets with four different original classifiers. Only the proposed algorithm and the algorithm of Hardt et al. (2016) eliminate bias in all cases whereas ROC can fail in $k$NN (e.g. ADULT dataset) because debiasing $k$NN can require randomization at the thresholds. Bias here is the absolute difference in mean outcomes across the sensitive attribute (Definition 1).

cannot be zero because randomization around the threshold is, in general, necessary for Bayes risk consistency as illustrated in the following example:

**Example 1** (Randomization is necessary). *Suppose that $\mathcal{X} = \{-1, 0, 1\}$ where $p(\boldsymbol{x} = -1) = 1/2$, $p(\boldsymbol{x} = 0) = 1/3$ and $p(\boldsymbol{x} = 1) = 1/6$. Let $\eta(-1) = 0$, $\eta(0) = 1/2$ and $\eta(1) = 1$. In addition, let $\boldsymbol{s} \in \{0, 1\}$ be a sensitive attribute, where $p(\boldsymbol{s} = 1|\boldsymbol{x} = -1) = 1/2$, $p(\boldsymbol{s} = 1|\boldsymbol{x} = 0) = 1$, and $p(\boldsymbol{s} = 1|\boldsymbol{x} = 1) = 0$. Then, the Bayes optimal prediction rule $f^\star(x)$ subject to statistical parity ($\epsilon = 0$) satisfies: $p(f^\star(\boldsymbol{x}) = 1|\boldsymbol{x} = -1) = 0$, $p(f^\star(\boldsymbol{x}) = 1|\boldsymbol{x} = 0) = 7/10$ and $p(f^\star(\boldsymbol{x}) = 1|\boldsymbol{x} = 1) = 1$.*

Note that the Bayes excess risk bound in Theorem 2 is vacuous when $\gamma = 0$. Therefore, $\gamma$ controls a trade-off depending on how crucial randomization is around the thresholds (e.g. in $k$-NN where the classifier's scores come from a finite set or in deep neural networks that tend to produce scores concentrated around $\{-1, +1\}$). In our experiments, $\gamma$ is always chosen using a validation set.

## 5 EMPIRICAL EVALUATION

**Experiment Setup** We compare against three post-processing methods: (1) the post-processing algorithm of Hardt et al. (2016) (2) the shift inference method, first introduced in (Saerens et al., 2002) and used more recently in (Wang et al., 2020), and (3) the Reject Option Classifier (ROC) (Kamiran et al., 2012). We use the implementation of the algorithm of Hardt et al. (2016) in the Fair-Learn software package (Dudik et al., 2020). The training data used for the post-processing methods is always a fresh sample, i.e. different from the data used to train the original classifiers. The value of the hyper-parameter $\theta$ of the ROC algorithm is chosen in the grid $\{0.01, 0.02, \ldots, 1.0\}$. When ROC fails, its solution with the minimum bias is reported. In the proposed algorithm, the parameter $\gamma$ is chosen in the grid $\{0.01, 0.02, 0.05, 0.1, 0.2, \ldots, 1.0\}$ while $\rho$ is chosen in the gird $\mathbb{E}[\mathbf{y}] \pm \{0, 0.05, 0.1\}$. All hyper-parameters are selected based on a separate validation dataset.

**Tabular Data** We empirically evaluate the performance of the proposed algorithm and the baselines on two real-world datasets, namely the Adult income dataset and the Default of Credit Card

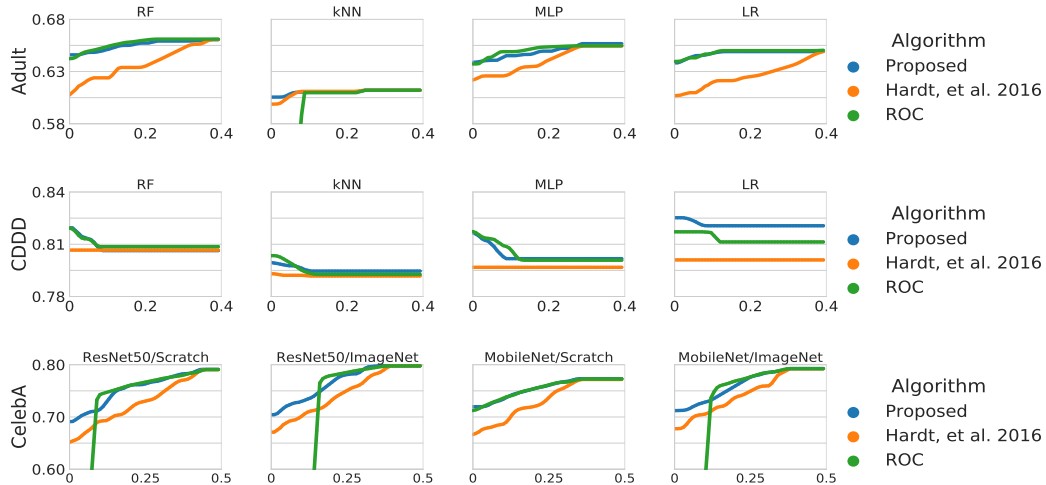

Figure 3: The tradeoff curves are displayed for each classification problem. The $x$-axis corresponds to bias (Definition 1) while the $y$-axis is the test accuracy. In general, debiasing CDDD improves test accuracy because bias was introduced to the training data only. In addition, ROC fails at debiasing four classifiers (see also Tables 1 and 2) due to the absence of randomization.

Clients (DCCC) dataset, both taken from the UCI Machine Learning Repository (Blake and Merz, 1998). The Adult dataset contains 48,842 records with 14 attributes each and the goal is to predict if the income of an individual exceeds $50K per year. The DCCC dataset contains 30,000 records with 24 attributes, and the goal is to predict if a client will default on their credit card payment. Both datasets include sensitive attributes, such as sex and age. In Figure 1 we showcased why, in some cases, the sensitive attribute can be the cross product of multiple features (e.g. religion, gender, and race). In our experiments in this section, we define the sensitive class to be the class of females. In the DCCC dataset, we additionally introduce bias in the training set for the purpose of the experiment: if $s(\mathbf{x}) = y(\mathbf{x})$ we keep the instance and otherwise drop it with probability 0.5.

We train four classifiers on each dataset: (1) random forests with maximum depth 10, (2) $k$-NN with $k = 10$, (3) a two-layer fully connected neural network with 128 hidden nodes, and (4) logistic regression. For the latter, we fine-tune the parameter $C$ in a grid of values chosen in a logarithmic scale between $10^{-4}$ and $10^4$ using 10-fold cross validation. The learning rate in our algorithm is fixed to $10^{-1}(K/T)^{1/2}$, where $T$ is the number of steps, and $\epsilon = 0$.

Table 1 shows the bias and accuracy on *test* data after applying each post-processing method. The column marked as "original" corresponds to the original classifier without any alteration. As shown in the table, both our proposed algorithm and the algorithm of Hardt et al. (2016) eliminate bias in all classifiers. By contrast, the shift-inference method does not succeed at controlling statistical parity while the ROC method can fail when the output of the original classifier is discrete, such as in $k$NN, because it does not learn to randomize. Moreover, the proposed algorithm has a much lower impact on the test accuracy compared to Hardt et al. (2016) and can even improve it in certain cases. The fact that fairness can sometimes improve accuracy was recently noted by Blum and Stangl (2020). The full tradeoff curves between bias and performance are provided in Figure 3.

**CelebA Dataset** Our second set of experiments builds on the task of predicting the "attractiveness" attribute in the CelebA dataset (Liu et al., 2015). CelebA contains 202,599 images of celebrities annotated with 40 binary attributes, including gender. We use two standard deep neural network architectures: ResNet50 (He et al., 2016) and MobileNet (Howard et al., 2017), trained from scratch or pretrained on ImageNet. We present the results in Table 2. We observe that the proposed algorithm significantly outperforms the post-processing algorithm of Hardt et al. (2016) and performs, at least, as well as the ROC algorithm whenever the latter algorithm succeeds. Often, however, ROC fails at debiasing the deep neural networks because it does not learn to randomize when most scores produced by neural networks are concentrated around the set $\{-1, +1\}$.

| | | Bias | | | | |
|---|---|---|---|---|---|---|
| Model | Training | Original | Proposed | Hardt et al. (2016) | Shift Inference | ROC |
| RESNET50 | SCRATCH | .43 | .01 | .01 | .38 | .08 |
| | IMAGENET | .40 | .02 | .01 | .35 | .15 |
| MOBILENET | SCRATCH | .35 | .01 | .01 | .24 | .01 |
| | IMAGENET | .38 | .002 | .002 | .34 | .10 |

| | | Test error | | | | |
|---|---|---|---|---|---|---|
| Model | Training | Original | Proposed | Hardt et al. (2016) | Shift Inference | ROC |
| RESNET50 | SCRATCH | 22.2% | 28.7% | 34.1% | 37.8% | 26.9% |
| | IMAGENET | 20.3% | 28.3% | 32.5% | 20.7% | 22.8% |
| MOBILENET | SCRATCH | 23.1% | 28.2% | 33.6% | 24.1% | 28.3% |
| | IMAGENET | 20.7% | 27.2% | 32.5% | 21.0% | 24.5% |

Table 2: A comparison of the four post-processing methods on CelebA (predict attractiveness) applied to the output of ResNet50 and MobileNet, each trained either from scratch or on ImageNet. The proposed algorithm performs much better than ROC in terms of bias and much better than Hardt et al. (2016) in terms of accuracy. Shift inference performs poorly in both objectives.

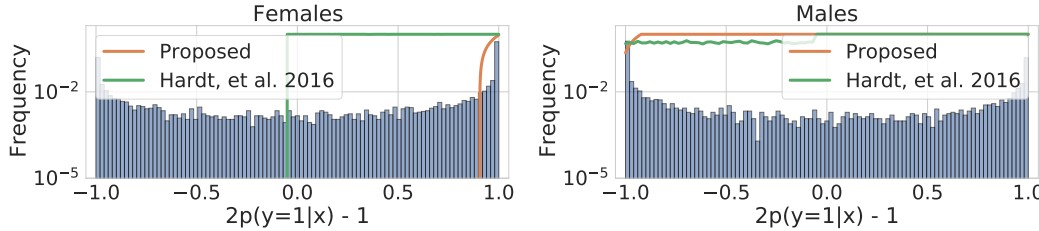

Figure 4: The distribution of the scores produced by ResNet50 trained from scratch are shown for both subpopulations. The curves correspond to the randomized post-processing rules, i.e. $p(\mathbf{y} = 1|\mathbf{x})$, of Hardt et al. (2016) and the proposed algorithm with $\gamma = 0.1$ and $\rho = \mathbb{E}[\mathbf{y}]$.

We investigated the strong performance compared to that of Hardt et al. (2016) and found that it is due to the specific form of randomization used by the proposed algorithm. As shown in Figure 4, the post-processing algorithm of Hardt et al. (2016) uses a fixed probability when randomizing between two thresholds. For CelebA trained from scratch, for example, the post-processing rule of Hardt et al. (2016) predicts nearly uniformly at random when ResNet50 predicts the negative class for males. In contrast, our algorithm uses a ramp function that takes the confidence of the scores into account. In Figure 4, in particular, the male instances with scores close to -1 are flipped with probability $\approx 0.15$, as opposed to $\approx 0.5$ in Hardt et al. (2016), and this difference is compensated for by flipping all examples with scores larger than $\approx -0.9$ and all female instances with scores less than $\approx 0.9$. Hence, less randomization is applied when the original classifier is more confident.

Lastly, one important observation we note in Table 2 is the impact of *pre-training* – pretraining in our experiments helps in achieving a lower test error rate even after eliminating bias. In other words, pretraining seems to reduce the cost of debiasing trained models.

## 6 CONCLUDING REMARKS

In this paper, we propose a near-optimal post-processing algorithm for debiasing trained machine learning models. The proposed algorithm is scalable, does not require retraining the classifiers, and has a limited impact on the test accuracy. In addition to providing strong theoretical guarantees, we show that it outperforms previous post-processing methods for unbiased classification on standard benchmarks across classical and modern machine learning models.

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

# A    FULL ALGORITHM

---

**Algorithm 1:** A Pseudocode of the Proposed Algorithm for Conditional Statistical Parity.

---

**Data:** $\gamma > 0$; $\rho \in [0, 1]$; $\epsilon \geq 0$; $f : \mathcal{X} \to [-1, +1]$; $s : \mathcal{X} \to [K]$

**Result:** Optimal values of thresholds: $(\lambda_1, \mu_1), \ldots, (\lambda_K, \mu_K)$.

**Training:** Initialize $(\lambda_1, \mu_1), \ldots, (\lambda_K, \mu_K)$ to zeros. Then, repeat until convergence:

1. Sample an instance $\mathbf{x} \sim p(x)$

2. Perform the updates:

$$\lambda_{s(\mathbf{x})} \leftarrow \max \left\{ 0, \ \lambda_{s(\mathbf{x})} - \eta \left( \frac{\epsilon}{2} + \rho + \frac{\partial}{\partial \lambda_{s(\mathbf{x})}} \xi_\gamma \left( f(\mathbf{x}) - (\lambda_{s(\mathbf{x})} - \mu_{s(\mathbf{x})}) \right) \right) \right\}$$

$$\mu_{s(\mathbf{x})} \leftarrow \max \left\{ 0, \ \mu_{s(\mathbf{x})} - \eta \left( \frac{\epsilon}{2} - \rho + \frac{\partial}{\partial \mu_{s(\mathbf{x})}} \xi_\gamma \left( f(\mathbf{x}) - (\lambda_{s(\mathbf{x})} - \mu_{s(\mathbf{x})}) \right) \right) \right\},$$

where $\xi_\gamma$ is given by Eq. (3).

**Prediction:** Given an instance $x$ in the group $X_k$, predict the label $+1$ with probability:

$$\tilde{h}(x) = \begin{cases} 0, & f(x) \leq \lambda_k - \mu_k \\ (f(x) - (\lambda_k - \mu_k))/\gamma, & \lambda_k - \mu_k \leq f(x) \leq \lambda_k - \mu_k + \gamma \\ 1, & f(x) \geq \lambda_k - \mu_k + \gamma \end{cases}$$

---

# B    PROOF OF THEOREM 1

## B.1    CONSTRAINED CONVEX FORMULATION

Suppose we have a binary classifier on the instance space $\mathcal{X}$. We would like to construct an algorithm for post-processing the predictions made by that classifier such that we control the bias with respect to a set of pairwise disjoint groups $X_1, \ldots, X_K \subseteq \mathcal{X}$ according to Definition 1. We assume that the output of the classifier $f : \mathcal{X} \to [-1, +1]$ is an estimate to $2\eta(x) - 1$, where $\eta(x) = p(\mathbf{y} = 1 | \mathbf{x} = x)$ is the Bayes regressor. This is not a strong assumption because many algorithms can be calibrated to provide probability scores (Platt et al., 1999; Guo et al., 2017) so the assumption is valid. We consider randomized rules of the form:

$$\tilde{h} : \mathcal{X} \times \{1, 2, \ldots, K\} \times [-1, 1] \to [0, 1],$$

whose arguments are: (1) the instance $\mathbf{x} \in \mathcal{X}$, (2) the sensitive attribute $s(x) \in [K]$, and (3) the original classifier's score $f(x)$. Because randomization is sometimes necessary as proved in Section 4, $\tilde{h}(x)$ is the probability of predicting the positive class when the instance is $x \in \mathcal{X}$.

Suppose we have a training sample of size $N$, which we will denote by $\mathcal{S}$. Let $q_i = \tilde{h}(x_i) \in [0, 1]$ for the $i$-th instance in $\mathcal{S}$. For each group $X_k \subseteq \mathcal{S}$, the fairness constraint in Definition 1 over the training sample can be written as:

$$\frac{1}{|X_k|} \left| \sum_{i \in X_k} q_i - \rho \right| \leq \frac{\epsilon}{2},$$

for some hyper-parameter $\rho > 0$. This holds by the triangle inequality.

To learn $\tilde{f}$, we propose solving the following *regularized* optimization problem:

$$\min_{0 \leq q_i \leq 1} \quad \sum_{i=1}^{N} (\gamma/2) \, q_i^2 \, - \, f(x_i) \, q_i \quad \text{s.t.} \quad \forall X_k \in \mathcal{G} : \left| \sum_{i \in X_k} q_i - \rho \right| \leq \epsilon_k \tag{6}$$

where $\gamma > 0$ is a regularization parameter and $\epsilon_k = |X_k| \, \epsilon/2$.

## B.2    REDUCTION TO UNCONSTRAINED OPTIMIZATION

Because the groups $X_k$ are pairwise disjoint, the optimization problems in (6) decomposes into $K$ separate suboptimization problems, one for each group $X_k$. Each sub-optimization problem can be

written in the following general form:

$$\min_{0 \le q_i \le 1} \sum_{i=1}^{M} \frac{\gamma}{2} q_i^2 - f(x_i) q_i$$

$$\text{s.t.} \qquad \sum_{i=1}^{M} (z_i q_i - b) \le \epsilon, \qquad -\sum_{i=1}^{M} (z_i q_i - b) \le \epsilon'$$

To recall, $\epsilon' = M\epsilon/2$. The Lagrangian is:

$$L(q, \alpha, \beta, \lambda, \mu) =$$
$$\sum_i \left( \frac{\gamma}{2} q_i^2 - f(x_i) q_i \right) + \lambda \left( \sum_i (z_i q_i - b) - \epsilon' \right) - \mu \left( \sum_i (z_i q_i - b) + \epsilon' \right) + \sum_i \alpha_i (q_i - 1) - \sum_i \beta_i q_i$$

Taking the derivative w.r.t. $q_i$ gives us:

$$q_i = \frac{1}{\gamma} \Big( f(x_i) - (\lambda - \mu) z_i - \alpha_i + \beta_i \Big)$$

Plugging this back, the dual problem becomes:

$$\min_{q, \lambda, \mu, \alpha, \beta} \sum_i \left( \frac{\gamma}{2} q_i^2 + b(\lambda - \mu) \right) + (\lambda + \mu)\epsilon' + \sum_i \alpha_i$$

$$\text{s.t.} \qquad q_i = \frac{1}{\gamma} \Big( f(x_i) - (\lambda - \mu) z_i - \alpha_i + \beta_i \Big)$$

$$\lambda, \mu, \alpha_i, \beta_i \ge 0$$

Next, we eliminate variables. By eliminating $\beta_i$, we have:

$$\min_{q, \lambda, \mu, \alpha, \beta} \sum_i \left( \frac{\gamma}{2} q_i^2 + b(\lambda - \mu) \right) + (\lambda + \mu)\epsilon' + \sum_i \alpha_i$$

$$\text{s.t.} \qquad q_i - \frac{1}{\gamma} \Big( f(x_i) - (\lambda - \mu) z_i - \alpha_i \Big) \ge 0$$

$$\lambda, \mu, \alpha_i \ge 0$$

Equivalently:

$$\min_{q, \lambda, \mu, \alpha, \beta} \sum_i \left( \frac{\gamma}{2} q_i^2 + b(\lambda - \mu) \right) + (\lambda + \mu)\epsilon' + \sum_i \alpha_i$$

$$\text{s.t.} \qquad \alpha_i \ge f(x_i) - \gamma q_i - (\lambda - \mu) z_i$$

$$\lambda, \mu, \alpha_i \ge 0$$

Next, we eliminate $\alpha_i$ to obtain:

$$\min_{q, \lambda, \mu} \sum_i \left( \frac{\gamma}{2} q_i^2 + b(\lambda - \mu) \right) + (\lambda + \mu)\epsilon' + \sum_i \left[ f(x_i) - \gamma q_i - (\lambda - \mu) z_i \right]^+$$

$$\lambda, \mu \ge 0$$

Finally, let's eliminate the $q_i$ variables. For a given optimal $\mu$ and $\lambda$, it is straightforward to observe that the minimizer $q^\star$ to $\gamma/2 q^2 + [w - \gamma q]^+$ must lie in the set $\{0, w/\gamma, 1\}$. In particular, if $w/\gamma \le 0$, then $q^\star = 0$. If $w/\gamma \ge 1$, then $q^\star = 1$. Note here that we make use of the fact that $\gamma > 0$.

So, the optimal value of $q^\star$ to $\gamma/2 q^2 + [w - \gamma q]^+$ is:

$$\xi_\gamma(w) = \begin{cases} 0 & \frac{w}{\gamma} \le 0 \\ \frac{w^2}{2\gamma} & 0 \le \frac{w}{\gamma} \le 1 \\ w - \frac{\gamma}{2} & \frac{w}{\gamma} \ge 1 \end{cases}$$

From this, the optimization problem reduces to:

$$\min_{\lambda, \mu \ge 0} \sum_{i=1}^{N} \Big( b(\lambda - \mu) + \epsilon'(\lambda + \mu) + \xi_\gamma(f(x_i) - (\lambda - \mu) z_i) \Big) \tag{7}$$

This is a differentiable objective function and can be solved quickly using the projected gradient descent method (Boyd and Mutapcic, 2008). The projection step here is taking the positive parts of $\lambda$ and $\mu$. This leads to the update rules in Algorithm 1.

What about the solution? Given $\lambda$ and $\mu$, the solution of $q_i$ is a minimizer to;

$$\frac{\gamma}{2}q_i^2 + \left[f(x_i) - \gamma q_i - (\lambda - \mu)z_i\right]^+$$

As stated earlier, the solution is:

$$q_i = \begin{cases} 0, & f(x_i) \leq (\lambda - \mu)z_i \\ (1/\gamma)(f(x_i) - (\lambda - \mu)z_i), & \gamma(\lambda - \mu)z_i \leq f(x_i) \leq (\lambda - \mu)z_i + \gamma \\ 1, & f(x_i) \geq (\lambda - \mu)z_i + \gamma \end{cases} \tag{8}$$

So, we have a ramp function. In the proposed algorithm, we have $z_i = 1$ and $b = \rho$ for all examples. This proves Theorem 1.

## C  PROOF OF THEOREM 2

### C.1  OPTIMAL UNBIASED PREDICTORS

We begin by proving the following result, which can be of independent interest.

**Theorem 3.** *Let $f^\star = \arg\min_{f:\mathcal{X}\to\{0,1\}} \mathbb{E}[\mathbb{I}\{f(\boldsymbol{x}) \neq \boldsymbol{y}\}]$ be the Bayes optimal decision rule subject to group-wise affine constraints of the form $\mathbb{E}[w_k(\boldsymbol{x}) \cdot f(\boldsymbol{x}) \mid \boldsymbol{x} \in X_k] = b_k$ for some fixed partition $\mathcal{X} = \cup_k X_k$. If $w_k : \mathcal{X} \to \mathbb{R}$ and $b_k \in \mathbb{R}$ are such that there exists a constant $c \in (0,1)$ in which $p(f(x) = 1) = c$ will satisfy all the affine constraints, then $f^\star$ satisfies $p(f^\star(x) = 1) = \mathbb{I}\{\eta(x) > t_k\} + \tau_k \mathbb{I}\{\eta(x) = t_k\}$, where $\eta(x) = p(\boldsymbol{y} = 1|\boldsymbol{x} = x)$ is the Bayes regressor, $t_k \in [0,1]$ is a threshold specific to the group $X_k \subseteq \mathcal{X}$, and $\tau_k \in [0,1]$.*

*Proof.* Minimizing the expected misclassification error rate of a classifier $f$ is equivalent to maximizing:

$$\mathbb{E}[f(\mathbf{x}) \cdot \mathbf{y} + (1 - f(\mathbf{x})) \cdot (1 - \mathbf{y})] = \mathbb{E}\left[\mathbb{E}[f(\mathbf{x}) \cdot \mathbf{y} + (1 - f(\mathbf{x})) \cdot (1 - \mathbf{y})] \mid \mathbf{x}\right]$$
$$= \mathbb{E}\left[\mathbb{E}[f(\mathbf{x}) \cdot (2\eta(\mathbf{x}) - 1)] \mid \mathbf{x}\right] + \mathbb{E}[1 - \eta(\mathbf{x})]$$

Hence, selecting $f$ that minimizes the misclassification error rate is equivalent to maximizing:

$$\mathbb{E}[f(\mathbf{x}) \cdot (2\eta(\mathbf{x}) - 1)] \tag{9}$$

Instead of maximizing this directly, we consider the regularized form first. Writing $g(x) = 2\eta(x) - 1$, the optimization problem is:

$$\min_{0 \leq f(x) \leq 1} (\gamma/2)\mathbb{E}[f(\mathbf{x})^2] - \mathbb{E}[f(\mathbf{x}) \cdot g(\mathbf{x})] \qquad \text{s.t.} \qquad \mathbb{E}[w(\mathbf{x}) \cdot f(\mathbf{x})] = b$$

Here, we focused on one subset $X_k$ because the optimization problem decomposes into $K$ separate optimization problems, one for each $X_k$. If there exists a constant $c \in (0,1)$ such that $f(x) = c$ satisfies all the equality constraints, then Slater's condition holds so strong duality holds (Boyd and Vandenberghe, 2004).

The Lagrangian is:

$$(\gamma/2)\mathbb{E}[f(\mathbf{x})^2] - \mathbb{E}[f(\mathbf{x}) \cdot g(\mathbf{x})] + \mu(\mathbb{E}[w(\mathbf{x}) \cdot f(\mathbf{x})] - b) + \mathbb{E}[\alpha(\mathbf{x})(f(\mathbf{x}) - 1)] - \mathbb{E}[\beta(\mathbf{x})f(\mathbf{x})],$$

where $\alpha(x), \beta(x) \geq 0$ and $\mu \in \mathbb{R}$ are the dual variables.

Taking the derivative w.r.t. the optimization variable $f(x)$ yields:

$$\gamma f(x) = g(x) - \mu\, w(x) - \alpha(x) + \beta(x) \tag{10}$$

Therefore, the dual problem becomes:

$$\max_{\alpha(x),\beta(x)\geq 0} -(2\gamma)^{-1} \mathbb{E}[(g(\mathbf{x}) - \mu\, w(\mathbf{x}) - \alpha(\mathbf{x}) + \beta(\mathbf{x}))^2] - b\mu - \mathbb{E}[\alpha(\mathbf{x})]$$

We use the substitution in Eq. (10) to rewrite it as:

$$\min_{\alpha(x),\beta(x)\geq 0} (\gamma/2)\,\mathbb{E}[f(\mathbf{x})^2] + b\mu + \mathbb{E}[\alpha(\mathbf{x})]$$

$$\text{s.t.}\forall x \in \mathcal{X} : \gamma f(x) = g(x) - \mu\,w(x) - \alpha(x) + \beta(x)$$

Next, we eliminate the multiplier $\beta(x)$ by replacing the equality constraint with an inequality:

$$\min_{\alpha(x)\geq 0} (\gamma/2)\,\mathbb{E}[f(\mathbf{x})^2] + b\mu + \mathbb{E}[\alpha(\mathbf{x})]$$

$$\text{s.t.}\forall x \in \mathcal{X} : g(x) - \gamma f(x) - \mu\,w(x) - \alpha(x) \leq 0$$

Finally, since $\alpha(x) \geq 0$ and $\alpha(x) \geq g(x) - \gamma f(x) - \mu w(x)$, the optimal solution is the minimizer to:

$$\min_{f:\mathcal{X}\to\mathbb{R}} (\gamma/2)\mathbb{E}[f(\mathbf{x})^2] + b\mu + \mathbb{E}[\max\{0,\, g(\mathbf{x}) - \gamma f(\mathbf{x}) - \mu w(\mathbf{x})\}]$$

Next, let $\mu^\star$ be the optimal solution of the dual variable $\mu$. Then, the optimization problem over $f$ decomposes into separate problems, one for each $x \in \mathcal{X}$. We have:

$$f(x) = \arg\min_{\tau\in\mathbb{R}} \left\{ (\gamma/2)\tau^2 + [g(x) - \gamma\tau - \mu^\star\,w(x)]^+ \right\}$$

Using the same argument in Appendix B, we deduce that $f(x)$ is of the form:

$$f(x) = \begin{cases} 0, & g(x) - \mu^\star\,w(x) \leq 0 \\ 1 & g(x) - \mu^\star\,w(x) \geq \gamma \\ (1/\gamma)\,(g(x) - \mu^\star\,w(x)) & \text{otherwise} \end{cases}$$

Finally, the statement of the theorem holds by taking the limit as $\gamma \to 0^+$. $\qquad\square$

## C.2 EXCESS RISK BOUND

In this section, we write $\mathcal{D}$ to denote the underlying probability distribution and write $\mathcal{S}$ to denote the uniform distribution over the training sample (a.k.a. empirical distribution).

The parameter $\rho$ stated in the theorem is given by:

$$\rho = (1/2)\left(\max_{k\in\{1,2,...,K\}} \mathbb{E}_{\mathbf{x}}[h^\star(\mathbf{x})\,|\,\mathbf{x} \in X_k] + \min_{k\in\{1,2,...,K\}} \mathbb{E}_{\mathbf{x}}[h^\star(\mathbf{x})\,|\,\mathbf{x} \in X_k]\right)$$

Note that, by definition, the optimal classifier $h^\star$ that satisfies $\epsilon$ statistical parity also satisfies the constraint in (6) with this choice of $\rho$. Hence, with this choice of $\rho$, $h^\star$ remains optimal among all possible classifiers.

Observe that the decision rule depends on $x$ only via $f(x) \in [-1,+1]$. Hence, we write $\mathbf{z} = f(\mathbf{x})$. Since the thresholds are learned based on a fresh sample of data, the random variables $\mathbf{z}_i$ are i.i.d. In light of Eq. 9, we would like to minimize the expectation of the loss $l(\tilde{h}_\gamma, \mathbf{x}) = -f(\mathbf{x}) \cdot \tilde{h}_\gamma(\mathbf{x}) = -\mathbf{z} \cdot q(\mathbf{z}) \doteq \zeta(\mathbf{z})$ for some function $q : [-1,+1] \to [0,1]$ of the form shown in 2(a). Note that $\zeta$ is $2(1 + 1/\gamma)$-Lipschitz continuous within the same group and sensitive class. This is because the thresholds are always in the interval $[-1-\gamma, 1+\gamma]$; otherwise moving beyond this interval would not change the decision rule.

Let $\tilde{\mathbf{h}}_\gamma$ be the decision rule learned by the algorithm. Using Corollary 5 in (Xu and Mannor, 2010), we conclude that with a probability of at least $1 - \delta$:

$$\left|\mathbb{E}_{\mathcal{D}}[l(\tilde{\mathbf{h}}_\gamma, \mathbf{x})] - \mathbb{E}_{\mathcal{S}}[l(\tilde{\mathbf{h}}_\gamma, \mathbf{x})]\right| \leq \inf_{R\geq 1}\left\{\left(\frac{4}{R}(1 + \frac{1}{\gamma})\right) + 2\sqrt{\frac{2(R+K)\log 2 + 2\log\frac{1}{\delta}}{N}}\right\} \qquad (11)$$

Here, we used the fact that the observations $f(\mathbf{x})$ are bounded in the domain $[-1,1]$ and that we can first partition the domain into groups $X_k$ ($K$ subsets) in addition to partitioning the interval $[-1,1]$ into $R$ smaller sub-intervals and using the Lipschitz constant. Choosing $R = N^{\frac{1}{3}}$ and simplifying gives us with a probability of at least $1 - \delta$:

$$\left|\mathbb{E}_{\mathcal{D}}[l(\tilde{\mathbf{h}}_\gamma, \mathbf{x})] - \mathbb{E}_{\mathcal{S}}[l(\tilde{\mathbf{h}}_\gamma, \mathbf{x})]\right| \leq \frac{4(2 + \frac{1}{\gamma})}{N^{\frac{1}{3}}} + 2\sqrt{\frac{2K + 2\log\frac{1}{\delta}}{N}}$$

The same bound also applies to the decision rule $\mathbf{h}_\gamma^\star$ that results from applying optimal threshold with width $\gamma > 0$ (here, "optimal" is with respect to the underlying distribution) because the $\epsilon$-cover (Definition 1 in (Xu and Mannor, 2010)) is independent of the choice of the thresholds. By the union bound, we have with a probability of at least $1 - \delta$, both of the following inequalities hold:

$$\left|\mathbb{E}_\mathcal{D}[l(\tilde{\mathbf{h}}_\gamma, \mathbf{x})] - \mathbb{E}_\mathcal{S}[l(\tilde{\mathbf{h}}_\gamma, \mathbf{x})]\right| \leq \frac{4(2 + \frac{1}{\gamma})}{N^{\frac{1}{3}}} + 2\sqrt{\frac{2K + 2\log\frac{2}{\delta}}{N}} \tag{12}$$

$$\left|\mathbb{E}_\mathcal{D}[l(\mathbf{h}_\gamma^\star, \mathbf{x})] - \mathbb{E}_\mathcal{S}[l(\mathbf{h}_\gamma^\star, \mathbf{x})]\right| \leq \frac{4(2 + \frac{1}{\gamma})}{N^{\frac{1}{3}}} + 2\sqrt{\frac{2K + 2\log\frac{2}{\delta}}{N}} \tag{13}$$

In particular:

$$\mathbb{E}_\mathcal{D}[l(\tilde{\mathbf{h}}_\gamma, \mathbf{x})] \leq \mathbb{E}_\mathcal{S}[l(\tilde{\mathbf{h}}_\gamma, \mathbf{x})] + \frac{4(2 + \frac{1}{\gamma})}{N^{\frac{1}{3}}} + 2\sqrt{\frac{2K + 2\log\frac{2}{\delta}}{N}}$$

$$\leq \mathbb{E}_\mathcal{S}[l(\mathbf{h}_\gamma^\star, \mathbf{x})] + \gamma + \frac{4(2 + \frac{1}{\gamma})}{N^{\frac{1}{3}}} + 2\sqrt{\frac{2K + 2\log\frac{2}{\delta}}{N}}$$

$$\leq \mathbb{E}_\mathcal{D}[l(\mathbf{h}_\gamma^\star, \mathbf{x})] + \gamma + \frac{8(2 + \frac{1}{\gamma})}{N^{\frac{1}{3}}} + 4\sqrt{\frac{2K + 2\log\frac{2}{\delta}}{N}}$$

The first inequality follows from Eq. (12). The second inequality follows from the fact that $\tilde{\mathbf{h}}_\gamma$ is an empirical risk minimizer to the regularized loss, where $\mathbb{E}[\tilde{f}(\mathbf{x})^2] \leq 1$ since $\tilde{f}(x) \in [0, 1]$. The last inequality follows from Eq. (13).

Finally, we know that the thresholding rule $\mathbf{h}_\gamma^\star$ with width $\gamma > 0$ is, by definition, a minimizer to:

$$(\gamma/2)\mathbb{E}[h(\mathbf{x})^2] - \mathbb{E}[h(\mathbf{x}) \cdot f(\mathbf{x})]$$

among all possible bounded functions $h : \mathcal{X} \to [0, 1]$ subject to the desired fairness constraints. Therefore, we have:

$$(\gamma/2)\mathbb{E}[\mathbf{h}_\gamma^\star(\mathbf{x})^2] - \mathbb{E}[\mathbf{h}_\gamma^\star(\mathbf{x}) \cdot f(\mathbf{x})] \leq (\gamma/2)\mathbb{E}[\mathbf{h}^\star(\mathbf{x})^2] - \mathbb{E}[\mathbf{h}^\star(\mathbf{x}) \cdot f(\mathbf{x})]$$

Hence:

$$\mathbb{E}[l(\mathbf{h}_\gamma^\star, \mathbf{x})] = -\mathbb{E}[\mathbf{h}_\gamma^\star(\mathbf{x}) \cdot f(\mathbf{x})] \leq \gamma + \mathbb{E}[l(\mathbf{h}^\star, \mathbf{x})]$$

This implies the desired bound:

$$\mathbb{E}_\mathcal{D}[l(\tilde{\mathbf{h}}_\gamma, \mathbf{x})] \leq \mathbb{E}_\mathcal{D}[l(\mathbf{h}^\star, \mathbf{x})] + 2\gamma + \frac{8(2 + \frac{1}{\gamma})}{N^{\frac{1}{3}}} + 4\sqrt{\frac{2K + 2\log\frac{2}{\delta}}{N}}$$

Therefore, we have consistency if $N \to \infty$, $\gamma \to 0^+$ and $\gamma N^{\frac{1}{3}} \to \infty$. For example, this holds if $\gamma = O(N^{-\frac{1}{6}})$.

So far, we have assumed that the output of the original classifier coincides with the Bayes regressor. If the original classifier is Bayes consistent, i.e. $\mathbb{E}[|2\eta(\mathbf{x}) - 1 - f(\mathbf{x})|] \to 0$ as $N \to \infty$, then we have Bayes consistency of the post-processing rule by the triangle inequality.

## D    PROOF OF PROPOSITION 1

*Proof.* Since $|\xi_\gamma'(w)| \leq 1$, the gradient at a point $\mathbf{x}$ during SGD has a square $\ell_2$-norm bounded by $||(1 + \rho + \epsilon)^2$ at all rounds. Following the proof steps of (Boyd and Mutapcic, 2008) and using the fact that projections are contraction mappings, one obtains:

$$\frac{1}{T}\sum_{t=1}^{T}\left(\mathbb{E}[F^{(t)}] - F(\mu^\star)\right) \leq \frac{||\mu^\star||_2^2 + ||\gamma^\star||_2^2 + (1 + \rho + \epsilon)^2 T\alpha^2}{2T\alpha}$$

$$= \frac{(1 + \rho + \epsilon)^2\alpha}{2} + \frac{||\mu^\star||_2^2 + ||\gamma^\star||_2^2}{2T\alpha}$$

By Jensen's inequality, we have $\frac{1}{T}\sum_{t=1}^{T}\mathbb{E}[F^{(t)}] \le \mathbb{E}[F(\bar{\mu})]$. Plugging this into the earlier results yields:

$$\mathbb{E}[\bar{F}] - F(\mu^{\star})) \le \frac{(1+\rho+\epsilon)^2\alpha}{2} + \frac{||\mu^{\star}||_2^2 + ||\gamma^{\star}||_2^2}{2T\alpha}$$

$\square$

# E    EXTENSION TO OTHER CRITERIA

## E.1    CONTROLLING THE COVARIANCE

The proposed algorithm can, sometimes, be adjusted to control bias according to other criteria as well besides statistical parity. For example, we demonstrate in this section how the proposed post-processing algorithm can be adjusted to control the covariance between the classifier's prediction and the sensitive attribute when both are binary random variables.

Let $\mathbf{a}, \mathbf{b}, \mathbf{c} \in \{0, 1\}$ be random variables. Let $C(\mathbf{a}, \mathbf{b}) \doteq \mathbb{E}[\mathbf{a} \cdot \mathbf{b}] - \mathbb{E}[\mathbf{a}] \cdot \mathbb{E}[\mathbf{b}]$ be their covariance, and $C(\mathbf{a}, \mathbf{b} \mid \mathbf{c})$ their covariance conditioned on $\mathbf{c}$:

$$C(\mathbf{a}, \mathbf{b} \mid \mathbf{c} = c) = \mathbb{E}[\mathbf{a} \cdot \mathbf{b} \mid \mathbf{c} = c] - \mathbb{E}[\mathbf{a} \mid \mathbf{c} = c] \cdot \mathbb{E}[\mathbf{b} \mid \mathbf{c} = c]. \tag{14}$$

Then, one possible criterion for measuring bias is to measure the conditional/unconditional covariance between the classifier's predictions and the sensitive attribute when both are binary random variables. Because the random variables are binary, it is straightforward to show that achieving zero covariance implies independence.

Suppose we have a binary classifier on the instance space $\mathcal{X}$. We would like to construct an algorithm for post-processing the predictions made by that classifier such that we guarantee $|C(f(\mathbf{x}), 1_S(\mathbf{x}) \mid \mathbf{x} \in X_k)| \le \epsilon$, where $\mathcal{X} = \cup_k X_k$ is a total partition of the instance space. Informally, this states that the fairness guarantee with respect to the senstiive attribute $1_S : \mathcal{X} \to \{0, 1\}$ holds within each subgroup $X_k$.

We assume, again, that the output of the classifier $f : \mathcal{X} \to [-1, +1]$ is an estimate to $2\eta(x) - 1$, where $\eta(x) = p(\mathbf{y} = 1|\mathbf{x} = x)$ is the Bayes regressor and consider randomized rules of the form:

$$\tilde{h} : \mathcal{X} \times \{0, 1\} \times \{1, 2, \ldots, K\} \times [-1, 1] \to [0, 1],$$

whose arguments are: (i) the instance $\mathbf{x} \in \mathcal{X}$, (ii) the sensitive attribute $1_S : \mathcal{X} \to \{0, 1\}$, (iii) the sub-group membership $k : \mathcal{X} \to [K]$, and (iv) the original classifier's score $f(x)$. Because randomization is sometimes necessary as proved in Section 4, $\tilde{h}(x)$ is the probability of predicting the positive class when the instance is $x \in \mathcal{X}$.

Suppose we have a training sample of size $N$, which we will denote by $\mathcal{S}$. Let $q_i = \tilde{h}(x_i) \in [0, 1]$ for the $i$-th instance in $\mathcal{S}$. For each group $X_k \subseteq \mathcal{S}$, the desired fairness constraint on the covariance can be written as:

$$\frac{1}{|X_k|} \Big| \sum_{i \in X_k} (1_S(i) - \rho_k)\, q_i \Big| \le \epsilon,$$

where $\rho_k = \mathbb{E}_{\mathbf{x}}[1_S(\mathbf{x}) \mid \mathbf{x} \in X_k]$. This is because:

$$\frac{1}{|X_k|} \sum_{i \in X_k} (1_S(i) - \rho_k)\, q_i = \frac{1}{|X_k|} \sum_{i \in X_k} 1_S(i)\, \tilde{f}(i) - \frac{\rho_k}{|X_k|} \sum_{i \in X_k} \tilde{f}(i)$$

$$= \mathbb{E}[1_S(\mathbf{x}) \cdot \tilde{f}(\mathbf{x}) \mid \mathbf{x} \in X_k] - \mathbb{E}[1_S(\mathbf{x}) \mid \mathbf{x} \in X_k] \cdot \mathbb{E}[\tilde{f}(\mathbf{x}) \mid \mathbf{x} \in X_k]$$

$$= C(\tilde{f}(\mathbf{x}), 1_S(\mathbf{x}) \mid \mathbf{x} \in X_k),$$

where the expectation is over the training sample. Therefore, in order to learn $\tilde{h}$, we solve the *regularized* optimization problem:

$$\min_{0 \le q_i \le 1} \sum_{i=1}^{N} (\gamma/2)\, q_i^2 - f(x_i)\, q_i \quad \text{s.t.} \quad \forall X_k \in \mathcal{G} : \Big| \sum_{i \in X_k} (1_S(i) - \rho_k)\, q_i \Big| \le \epsilon_k \tag{15}$$

where $\gamma > 0$ is a regularization parameter and $\epsilon_k = |X_k|\, \epsilon$. This is of the same general form analyzed in Section B.2. Hence, the same algorithm can be applied with $b = 0$ and $z_i = 1_S(i) - \rho_k$.

### E.2   IMPOSSIBILITY RESULT

The previous algorithm for controlling covariance requires that the subgroups $X_k$ be known in advance. Indeed, our next impossibility result shows that this is, in general, necessary. In other words, a deterministic classifier $f : \mathcal{X} \to \{0,1\}$ cannot be universally unbiased with respect to a sensitive class $S$ across all possible known and unknown groups unless the representation $\mathbf{x}$ has zero mutual information with the sensitive attribute or if $f$ is constant almost everywhere. As a corollary, the groups $X_k$ have to be known *in advance*.

**Proposition 2** (Impossibility result). *Let $\mathcal{X}$ be the instance space and $\mathcal{Y} = \{0, 1\}$ be a target set. Let $1_S : \mathcal{X} \to \{0,1\}$ be an arbitrary (possibly randomized) binary-valued function on $\mathcal{X}$ and define $\gamma : \mathcal{X} \to [0,1]$ by $\gamma(x) = p(1_S(\mathbf{x}) = 1 \,|\, \mathbf{x} = x)$, where the probability is evaluated over the randomness of $1_S : \mathcal{X} \to \{0,1\}$. Write $\bar{\gamma} = \mathbb{E}_{\mathbf{x}}[\gamma(\mathbf{x})]$. Then, for any binary predictor $f : \mathcal{X} \to \{0,1\}$ it holds that*

$$\sup_{\pi : \mathcal{X} \to \{0,1\}} \left\{ \mathbb{E}_{\pi(\mathbf{x})} \left| \mathcal{C}(f(\mathbf{x}), \gamma(\mathbf{x}) | \, \pi(\mathbf{x})) | \right| \right\} \geq \frac{1}{2} \, \mathbb{E}_{\mathbf{x}} |\gamma(\mathbf{x}) - \bar{\gamma}| \cdot \min\{\mathbb{E}f, 1 - \mathbb{E}f\}, \quad (16)$$

*where $\mathcal{C}(f(\mathbf{x}), \gamma(\mathbf{x}) | \, \pi(\mathbf{x}))$ is defined in Equation 14.*

*Proof.* Fix $0 < \beta < 1$ and consider the subset:
$$W = \{x \in \mathcal{X} : \; (\gamma(x) - \bar{\gamma}) \cdot (f(x) - \beta) > 0\},$$
and its complement $\bar{W} = \mathcal{X} \setminus W$. Since $f(x) \in \{0,1\}$, the sets $W$ and $\bar{W}$ are independent of $\beta$ as long as it remains in the open interval $(0, 1)$. More precisely:
$$W = \begin{cases} \gamma(x) - \bar{\gamma} > 0 & \wedge & f(x) = 1 \\ \gamma(x) - \bar{\gamma} \leq 0 & \wedge & f(x) = 0 \end{cases}$$

Now, for any set $X \subseteq \mathcal{X}$, let $p_X$ be the projection of the probability measure $p(x)$ on the set $X$ (i.e. $p_X(x) = p(x)/p(X)$). Then, with a simple algebraic manipulation, one has the identity:
$$\mathbb{E}_{\mathbf{x} \sim p_X}[(\gamma(\mathbf{x}) - \bar{\gamma})(f(\mathbf{x}) - \beta)] = C(\gamma(\mathbf{x}), f(\mathbf{x}); \mathbf{x} \in X) + (\mathbb{E}_{\mathbf{x} \sim p_X}[\gamma] - \bar{\gamma}) \cdot (\mathbb{E}_{\mathbf{x} \sim p_X}[f] - \beta) \tag{17}$$

By definition of $W$, we have:
$$\mathbb{E}_{\mathbf{x} \sim p_W}[(\gamma(\mathbf{x}) - \bar{\gamma})(f(\mathbf{x}) - \beta)] = \mathbb{E}_{\mathbf{x} \sim p_W}[|\gamma(\mathbf{x}) - \bar{\gamma}||f(\mathbf{x}) - \beta|] \geq \min\{\beta, 1 - \beta\} \mathbb{E}_{\mathbf{x} \sim p_W} |\gamma(\mathbf{x}) - \bar{\gamma}|$$

Combining this with Eq. (17), we have:
$$C(\gamma(\mathbf{x}), f(\mathbf{x}); \mathbf{x} \in W) \geq \min\{\beta, 1 - \beta\} \mathbb{E}_{\mathbf{x} \sim p_W} |\gamma(\mathbf{x}) - \bar{\gamma}| + (\mathbb{E}_{\mathbf{x} \sim p_W}[\gamma] - \bar{\gamma})(\beta - \mathbb{E}_{\mathbf{x} \sim p_W}[f]) \tag{18}$$
Since the set $W$ does not change when $\beta$ is varied in the open interval $(0, 1)$, the lower bound holds for any value of $\beta \in (0, 1)$. W set:
$$\beta = \bar{f} \doteq \frac{1}{2} \left( \mathbb{E}_{\mathbf{x} \sim p_W} f(\mathbf{x}) + \mathbb{E}_{\mathbf{x} \sim p_{\bar{W}}} f(\mathbf{x}) \right) \tag{19}$$
Substituting the last equation into Eq. (18) gives the lower bound:
$$C(\gamma(\mathbf{x}), f(\mathbf{x}); \mathbf{x} \in W) \geq \min\{\bar{f}, 1 - \bar{f}\} \cdot \mathbb{E}_{\mathbf{x} \sim p_W} |\gamma(\mathbf{x}) - \bar{\gamma}|$$
$$+ \frac{1}{2}(\mathbb{E}_{\mathbf{x} \sim p_W}[\gamma] - \bar{\gamma})(\mathbb{E}_{\mathbf{x} \sim p_W} f(\mathbf{x}) - \mathbb{E}_{\mathbf{x} \sim p_{\bar{W}}} f(\mathbf{x})) \tag{20}$$
Repeating the same analysis for the subset $\bar{W}$, we arrive at the inequality:
$$C(\gamma(\mathbf{x}), f(\mathbf{x}); \mathbf{x} \in \bar{W}) \leq - \min\{\bar{f}, 1 - \bar{f}\} \, \mathbb{E}_{\mathbf{x} \sim p_{\bar{W}}} |\gamma(\mathbf{x}) - \bar{\gamma}|$$
$$+ \frac{1}{2}(\mathbb{E}_{\mathbf{x} \sim p_{\bar{W}}}[\gamma] - \bar{\gamma})(\mathbb{E}_{\mathbf{x} \sim p_W} f(\mathbf{x}) - \mathbb{E}_{\mathbf{x} \sim p_{\bar{W}}} f(\mathbf{x})) \tag{21}$$
Writing $\pi(x) = 1_W(x)$, we have by the reverse triangle inequality:
$$\mathbb{E}_{\pi(\mathbf{x})} \left| \mathcal{C}(f(\mathbf{x}), \gamma(\mathbf{x}); \, \pi(\mathbf{x})) \right| \geq \min\{\bar{f}, 1 - \bar{f}\} \cdot \mathbb{E}_{\mathbf{x}} |\gamma(\mathbf{x}) - \bar{\gamma}| \tag{22}$$
Finally:
$$2\bar{f} \geq p(\mathbf{x} \in W) \cdot \mathbb{E}_{\mathbf{x} \sim p_W} f(\mathbf{x}) + p(\mathbf{x} \in \bar{W}) \cdot \mathbb{E}_{\mathbf{x} \sim p_{\bar{W}}} f(\mathbf{x}) = \mathbb{E}[f]$$
Similarly, we have $2(1 - \bar{f}) \geq 1 - \mathbb{E}[f]$. Therefore:
$$\min\{\bar{f}, 1 - \bar{f}\} \geq \frac{1}{2} \min\{\mathbb{E}f, 1 - \mathbb{E}f\}$$
Combining this with Eq. (22) establishes the statement of the proposition. $\qquad \square$

