# OpenReview forum: "A Near-Optimal Recipe for Debiasing Trained Machine Learning Models"
_ICLR.cc/2021/Conference — Reject_

### Official Review · AnonReviewer1 · 2020-10-23
**weak paper on an important subject**

**Rating:** 4
**Confidence:** 4

**Review:**

Evaluation and improvement of fairness of machine learning algorithms is a very important issue. To this end, the authors of this paper propose a post-processing algorithm to enforce fairness in a narrowly defined notion of fairness. Unfortunately, I have serious concerns about the validity of the results and conclusions of the paper, and hence I cannot recommend the paper to be accepted.

[**Definition**] This entire paper is only applicable to a narrow definition of fairness in Definition 1. Given that $a$ and $b$ are binary, Definition 1 (page 2) is essentially equivalent to assuming that $P[a= b =1 |c] = P[a= 1 |c]  P[b= 1 |c]$ which is weaker than the well-known demographic (statistical) parity, and is only applicable to a binary setting. Hence, the applicability of the proposed algorithm is extremely limited.

[**Theory**] The theoretical exposure in this paper is confusing and not rigorous. The trouble begins from the unnumbered equation in page 4, where the authors define $\widetilde{h}$. It is unclear what $\widetilde{h}$ depends on. Also, the output is seemingly binary, but the authors claim this a _randomized prediction rule_ and $\widetilde{h}(\mathbf{x})$ represents the probability of predicting the positive class. **Proposition 1** is an obvious statement and not relevant to the claims of this paper, and hence should be removed. **Theorem 1:** In the proof of Theorem 1, the authors make an assumption about what the output of $f$ is. In different places the output is assumed to be in $[0,1]$ and $[-1, 1]$!! Even worse, they claim that the empirical average of some loss value is equal to its expectation (_what happened to generalization?_). That is where I stopped reading.

[**Experiments**] The experiments are very weak and not convincing. The paper only compares with Hardt et al. but in an unconvincing way as detailed here. **(a)** the comparisons should be made in terms of a tradeoff curve between _fairness_ and _performance_. In the currently reported results, there are instances where Hard et al. and the proposed method are not comparable, e.g., kNN. **(b)** The comparisons are unfair because they are done with respect to Definition 1, which is a very narrow sense of fairness while Hard et al. impose fairness either with respect to _equalized odds_ or _equal opportiunity_  which require conditional independence of $\widehat{Y}$ and $S$ given $Y (=1)$. Hence, the comparison with Hard et al. is unjustified. At the very least the authors should compare with a plethora of baselines designed for demographic parity (e.g., Kamiran et al. 2012, Zemel et al. 2013, Feldman et al. 2015, Zafar et al. 2017, Jiang et al. 2019, Baharlouei at al. 2020). Even then, the comparison should be done with respect to the statistical parity violation as defined in Dwork et al. 2012, which is a more established notion of fairness.

===== after rebuttal =====

I'd like to thank the authors for their revisions, which have significantly improved the readability of the paper, and the presentation of the results. The addition of fairness violation/accuracy tradeoffs and also (Kamiran et al. 2012) add a lot of value in putting this paper in the right context. After reading the rebuttal, I still remain unconvinced about the contributions of this paper. From a practical point of view, the performance of the proposed algorithm is on a par with (Kamiran et al. 2012), which is a baseline for demographic parity and has been improved on several times in the past 8 years. On the other hand, the main claim of the paper seems to be theoretical optimality. Unfortunately, although several previous mistakes have been corrected in the proofs, I cannot still follow the proofs of Theorems 1 and 2. New notation pops up all over the proof, and it is unclear how to follow some lines from the others. Given this, I am increasing my rating to 4 in acknowledgement of my previous misunderstanding of the definition of the statistical parity, which was cleared by the authors during the discussion period. However, I am still unable to recommend this paper in the current form for publication in the conference proceedings.

---

> ### Author Response · Authors · 2020-11-12
> **Response to reviewer**
>
> Thank you for your comments. Please see our response below.
>
> [Statistical parity] We disagree with the assessment that conditional statistical parity is “narrow” as it is a strict generalization to “statistical parity”, which as you mentioned, is a well-established notion of fairness used in the literature. In fact, a recent crowdsourcing study has found that it most closely matches with the human perception of bias [1]. As such, it has gathered a lot of interest in the community (e.g. [2], [3]).
>
> [Binary vs non-binary] This binary case is presented in the paper for clarity and it is straightforward to extend the algorithm to non-binary sensitive attributes. We have included an outline of how the algorithm can be extended to non-binary sensitive attributes in Appendix F (please see the revised version).
>
> [Proposition 1] The reason we include it is to clarify why the algorithm assumes that the groups X_k are known in advance as this was a common question and the proposition shows that it is indeed necessary. We disagree that the explicit lower bound in Proposition 1 is obvious! In general, we agree that this is non-central to the rest of the paper and we can defer it to the appendix if needed.
>
> [Theorem 1]  We use $f(x)$ for the output of the original classifier, which is assumed to be in [-1, +1] and we used $\tilde f(x)$ for the new rule learned by the post-processing algorithm whose range is [0, 1]. There is no contradiction since these are different functions. We have replaced $\tilde f(x)$ with $\tilde h(x)$ to avoid such confusion. Regarding the question about generalization, the claim of Theorem 1 is that the algorithm satisfies the desired fairness guarantee on the training sample. The fairness guarantee on the population (test data) is provided by Theorem 2. We have clarified this in the statement of Theorem 1 (please see the revised version).
>
> [Empirical validation] We propose a post-processing algorithm and we show that randomization is key (both in theory and in practice). The closest algorithm to ours with a randomized rule is the post-processing algorithm by Hardt et, al. (2016), which can be used for statistical parity (see for instance the implementation available in the FairLearn software package by Dudik et al., 2020). To clarify, our claim is not that our algorithm subsumes (Hardt et, al. 2016) because the latter can be used for equalized odds and equality of opportunity as well. Rather, our claim is that for statistical parity, or more generally conditional statistical parity, our algorithm significantly outperforms it. Regarding kNN, we also report results on 3 other classical algorithms (logistic regression, random forests, and MLP) as well as results on modern state-of-the-art neural network architectures (ResNet50 and MobileNet) both trained from scratch and pre-trained on ImageNet. Our conclusions hold across all settings, not just in kNN. As for the other algorithms, we propose a post-processing algorithm so we focused on post-processing rules. Kamiran et al. 2012 is a preprocessing algorithm, Zemel et al. 2013 is a representation learning method, and so on. We do believe post-processing is advantageous, particularly for deep learning, for the reasons mentioned in the paper.
>
> [Typos] Regarding the unnumbered equation in Page 4, thanks for catching that typo: The range is [0, 1], not {0, 1}. This has been fixed in the revised version. Please note that we have explicitly mentioned what h depends on right below the equation.
>
> [1] Srivastava, Megha, Hoda Heidari, and Andreas Krause. "Mathematical notions vs. human perception of fairness: A descriptive approach to fairness for machine learning." In SIGKDD, 2019.
> [2] S. Corbett-Davies, E. Pierson, A. Feller, S. Goel, and A. Huq, “Algorithmic decision making
> and the cost of fairness,” in Proceedings of the 23rd ACM SIGKDD International Conference
> on Knowledge Discovery and Data Mining, 2017, pp. 797–806.
> [3] N. Mehrabi, F. Morstatter, N. Saxena, K. Lerman, and A. Galstyan, “A survey on bias and
> fairness in machine learning,” arXiv preprint arXiv:1908.09635, 2019

---

> > ### Comment · AnonReviewer1 · 2020-11-12
> > **Notion of fairness and comparison metric**
> >
> > Thanks for the quick response. I just quickly respond to some points you bring up here before reading in detail the changes you have made to the paper.
> >
> > **Statistical parity**: Your notion of statistical parity in Definition 1 is strictly weaker than the one put forth by Dwork et al. 2012 because yours only requires that $P[a=b=1|c] = P[a=1|c]P[b=1|c]$ for $c \in$ {0, 1}, and leaves out other possibilities for $a$ and $b$, i.e., (0,1), (1,0), and (0,0). Hence, it is strictly weaker notion of fairness compared to the well-known statistical parity. You are welcome to use this notion to develop an algorithm but I find this notion unacceptable as a notion of fairness for evaluation and you provide no compelling argument otherwise.
> >
> > **Comparison metric**: Unless you compare with prior art using well defined notions of fairness, i.e., either **demographic parity** (as defined by Dwork et al.) or **equalized odds** or **equal opportunity**, your comparisons would not be acceptable. I understand that some of the other baselines in fairness are pre-processing or in-processing algorithms, and it is perfectly fine to compare with them as is customary in the literature. When you decide your fairness metric from the three defined above, you should compare with algorithms that are designed for the specific metric. For example, Hard et al. is specifically designed for **equalized odds** or **equal opportunity**, hence you are welcome to compare with them on those metrics.  If you choose to compare on **demographic parity**, you should compare with the algorithms that I mentioned which are specifically designed for that metric.
> >
> > **Comparison tradeoff**: Finally, in all your comparisons, you should report a tradeoff plot with one axis representing performance and the other representing fairness metric. Hard et al. is able to trade off performance for fairness and achieve a tradeoff curve. You should compare your own with theirs. It is not okay to just report one (fairness, performance) pair which is only a single point on the tradeoff curve.

---

> > > ### Author Response · Authors · 2020-11-13
> > > **Response**
> > >
> > > [Statistical Parity] Thanks for your comments. Regarding the notion of fairness, please note that for binary random variables, it is well-known that zero covariance implies independence. So, all of the other cases you mentioned (e.g.  (0,1),  (1,0),  (0,0)) are automatically accounted for. Here is a quick proof.
> > >
> > > Suppose that for two binary random variables $x,y\in$ {0,1}, our criteria is satisfied; i.e. $p(x=1,y=1)=p(x=1)p(y=1)$. We can show that this implies $x$ and $y$ are independent. First, we note that our condition implies $p(x=1) = p(x=1|y=1)$. By the sum rule, we have:
> > > \begin{align}
> > > p(x=1)=p(y=1)\cdot p(x=1|y=1)+p(y=0)\cdot p(x=1|y=0)\\
> > > =p(y=1)\cdot p(x=1)+p(y=0)\cdot p(x=1|y=0),
> > > \end{align}
> > > where we used the fact that $p(x=1) = p(x=1|y=1)$.
> > >
> > > Therefore, by rearranging terms: $p(x=1|y=0) = \frac{(1-p(y=1))p(x=1)}{p(y=0)} = p(x=1)$, which holds because $y$ is binary.
> > > Hence, we also have $p(x=1, y=0) = p(x=1)\cdot p(y=0)$. By symmetry, this also implies that $p(x=0, y=1) = p(x=0)\cdot p(y=1)$.
> > >
> > > Finally:
> > > \begin{align}
> > > p(x=0, y=0)= 1- p(x=1,y=1)-p(x=0,y=1)-p(x=1,y=0) \\
> > >     = 1-p(x=1)p(y=1)-p(x=0)p(y=1)-p(x=1)p(y=0)\\
> > >     = p(x=0)p(y=0),
> > > \end{align}
> > >
> > > where we used the previous result and the fact that all probabilities sum to 1.
> > > So, the two random variables are independent. This is why the condition  $p(x=1, y=1)=p(x=1)\cdot p(y=1)$ is sufficient.
> > >
> > > The definition of demographic parity states that $p(y=1|s)=p(y=1)$, i.e. $y$ and $s$ are independent of each other. As shown above, in the absence of groups $X_k$, our definition is equivalent to demographic parity. This is why we use a more general definition, not narrower.
> > >
> > > [Comparisons] Regarding the use of the post-processing algorithm of Hardt et, al. (2016), please note that the algorithm is a post-processing rule and can accommodate different types of constraints including demographic parity (see for instance Section 7 in the original paper (https://ttic.uchicago.edu/~nati/Publications/HardtPriceSrebro2016.pdf) as well as the implementation in the FairLearn package (https://fairlearn.github.io/v0.5.0/user_guide/mitigation.html#postprocessing). Also, we would like to refer you to the work of (Agrawal, et al. 2018) that used the algorithm for demographic parity in their comparison (http://proceedings.mlr.press/v80/agarwal18a/agarwal18a.pdf). In Section 4, second paragraph, the authors say: “This post-processing algorithm works with both demographic parity and equalized odds, as well as with binary and non-binary protected attributes.”
> > >
> > >
> > > [Comparison Tradeoff] For the tradeoff curves, please note that we did not select an arbitrary random point. We reported the performance when bias is eliminated. In all cases, our algorithm performs at least equally well in terms of bias, but much better in terms of accuracy. However, our algorithm also contains a hyper-parameter $\epsilon$ that allows one to trade off fairness for performance so we will work on generating the tradeoff curves as requested and include them in the appendix.

---

> > > > ### Comment · AnonReviewer1 · 2020-11-14
> > > > **Notion of fairness and comparisons -- ctd.**
> > > >
> > > > **Statistical Parity** Thanks for the explanation. I am convinced that your notion of statistical parity is equivalent to conditional independence. Please add a note to the paper stating that. However, your notion is still an odd notion of conditional independence between the predicted target and the sensitive attribute given the features. The existing notions are *demographic parity,* i.e., independence between predicted targets and sensitive attributes, and *equalized odds,* i.e., independence between sensitive attributes and the predicted target given the true target. Your notion amounts to neither of those and I remain unconvinced that it is a good notion of fairness.
> > > >
> > > > **Statistical Parity violation** Next, the existing metrics for measuring the amount of violation in fairness notions are based on the max violation over the set of possible values of the joint distribution. Yours is a different (and weaker) notion by considering that for (1,1) and ignoring the rest. I'd urge you to make sure that you report the violations in equalized odds or demographic parity the same way they are defined in the literature and widely used and refrain from comparing with existing art in the new notion.
> > > >
> > > > **Comparisons** Finally, these existing algorithms are all open-sourced and widely available, so there is no excuse in not reporting all of them and sticking with Hard et al., which is outdated at this point. Also, the comparison tradeoffs curves belong to the main body and not the appendix.

---

> > ### Comment · AnonReviewer1 · 2020-11-19
> > **More comments/questions**
> >
> > [binary vs. non-binary]
> > Thanks for adding Appendix F.
> > Can you please explain what $q_i$ is in (22) and how it is related to (21)?
> > Can you please showcase this solution through an experiment?
> >
> > [Proposition 1]
> > Yes, please move to appendix and add a one-line reference to it when you assume X_k's are known in advance.
> >
> > [Comparison]
> > Please provide a comparison with existing pre-processing, in-processing, and post-processing algorithms whenever possible. It is understood that post-processing might be preferred in some applications, but it is imperative to first establish the performance of your algorithm.
> >
> > [Theory]
> > Thanks for the explanations and corrections. I will be re-reading the theoretical results in the next few days, and will get back with more comments.

---

### Official Review · AnonReviewer4 · 2020-10-29
**Overall, I vote for accepting. Existing post-processing algorithms usually lack theoretical guarantees but not this paper. My main concern is the clarity.**

**Rating:** 6
**Confidence:** 4

**Review:**

The paper considers the fair classification problem with respect to conditional statistical parity. It proposes a near-optimal post-processing algorithm for debiasing trained machine learning models, including deep neural networks. The empirical results show that the algorithm outperforms existing post-processing approaches for fair classification.

Overall, I vote for accepting. Existing post-processing algorithms usually lack theoretical guarantees but not this paper. My main concern is the clarity; see cons.

Pros:
    1. The paper provides provable guarantees from both the impossible side and the algorithmic side.
    2. The post-processing approach does not require retraining the classifiers, and the impact on the test accuracy is limited.

Cons:
    1. Several symbols or notions lack explanations. E.g., what $\gamma$ means in Eq. (1)? What $\eta$ means in Eq. (2)?
    2. Theorems lack explanations. E.g., in Theorem 2, it is better to explain each term of the right side, including why these terms exist and when they are small.
    3. Equations lack explanations. E.g., why the update rules should be formulated as (2)? It is better to explain the intuition.

---

> ### Author Response · Authors · 2020-11-12
> **Response to reviewer**
>
> Thank you for your comments. We strived to incorporate your feedback and provide as much detail as possible given the limited space. We deferred the remaining discussion to the appendix. Specifically:
>
> - $\gamma$ is the regularization parameter that controls the width of randomization (see for example Figure 2(a)). It is introduced in Section 3. More details about it are available in Appendix C.1 (e.g. the discussion around Eq 13).
> -  We reiterated the definition of $\eta$ in Theorem 2 for clarity. Please see the revised version.
> - Regarding the right-hand side of Theorem 2, the first term is the Bayes risk which cannot be avoided. The second term involving gamma is due to the fact that the algorithm optimizes a regularized loss. The third and fourth term are due to the finite sample size and Lipschitz continuity of the loss. We specifically used the robustness-based framework introduced by Xu and Mannor (2010) to obtain that bound as mentioned in the paper.
> The update rules are derived in Appendix C. They correspond to the application of the projected SGD method.

---

### Official Review · AnonReviewer2 · 2020-10-30
**Well written. Timely.**

**Rating:** 7
**Confidence:** 4

**Review:**

In this paper, the authors propose a post-processing method for removing bias from a trained model. The bias is defined as conditional statistical parity — for a given partitioning of the data, the predicted label should be conditionally uncorrelated with the sensitive (bias inducing) attribute for each partition. The authors relax this strong requirement to an epsilon-constraint on the conditional covariance for each partition. As an example, race (sensitive attribute) should be conditionally uncorrelated to whether an individual will default on their loan (predicted target) for each city (data partition). The authors propose a constrained optimization problem that takes the input data, sensitive attribute, partitioning and a trained model to yield a probabilistic decision rule. Subsequently, they propose an iterative solution to the problem, proving some theoretical properties as well as showing how the method compares to different baselines.

Fairness is an important consideration as machine learning models find purchase in sensitive applications like loan approvals, job candidate filtering, compensation decisions, and so on. This clearly written paper summarizes the different ways of removing data bias, and proposes a sensible and fairly general post-processing solution. The paper is easy to follow, and while the main body skims on some details to assist readability (and adhere to the page limit), the appendix, which I only quickly scanned, is well-done. I did have some concerns:

1. The assumption that “the original classifier outputs a monotone transformation of some approximation to the posterior probability p(y = 1 | x)” needs further justification since models often tend to be overconfident of the wrong predictions violating the monotonicity assumption [Guo, Chuan, et al. "On calibration of modern neural networks." arXiv preprint arXiv:1706.04599 (2017)]. How central is this assumption to the analysis?

2. In section 3 (Algorithm subsection) doesn’t \tilde{h} represent probability? If so, the function range should be [0,1], and not {0,1}.

3. Right below, a bar is missing in the absolute value: “can be written as |.| < \epsilon”.

4. Theorem 2 notation: h(x) \neq y would be more cleanly written with an indication function: I(h(x) \neq y).

5. In baseline comparison, using only conditional statistical parity might be unfair to other methods which don’t optimize this notion of bias explicitly. Did you consider evaluating for other metrics?

Overall, this is a well-written paper that tackles an important problem, and it would make for a good addition to the conference.

---

> ### Author Response · Authors · 2020-11-12
> **Response to reviewer**
>
> Thank you for your comments. Please see our response below.
>
> [Assumptions] The assumption that “the original classifier outputs a monotone transformation of the Bayes regressor” is there only to motivate the development of the algorithm. It is by no means required -- both in theory (not necessary for the correctness in Theorem 1) and in practice (validated empirically). In addition, the algorithm performs well for a wide range of classifiers.
>
>
> [Theorem 2] We have included the indicator function as you suggested.
>
> [Empirical validation] Regarding the experiments, the algorithm we propose is specifically designed for conditional statistical parity, including the more commonly used demographic parity. Our experiments in Section 5 focus only on demographic parity in its unconditional form for the reasons you have mentioned, but the algorithm handles conditional statistical parity, in general, as shown in the experiment in Figure 1.
>
> [Typos] Thanks for pointing out the typos regarding the range of \tilde h and the missing bar in the constraint. They have been corrected (please see the revised version).

---

### Author Response · Authors · 2020-11-23
**Summary of Revision**

We would like to thank the reviewers for their careful reviews and useful suggestions which ultimately improved the manuscript. We have since revised the manuscript to address the specific comments:

* During discussions with Reviewer1, we concluded that instead of the covariance-based definition of statistical parity, we can instead focus the exposition on the criterion of maximum difference in mean outcomes, as it is easier to connect with related work. We note that both theoretical and empirical conclusions remain essentially the same. In addition, we have extended the algorithm to handle non-binary sensitive attributes. We moved the original covariance-based definition of bias (conditional statistical parity) and its related impossibility result to Appendix E to highlight that the algorithm can accommodate such a criteria as well.

* Instead of reporting a single point on the bias/accuracy tradeoff (the point with minimum bias), we now include the full tradeoff curves which contain more information (cf. Figure 3).

* We included an additional post-processing method, namely the Reject Option Classifier (Kamiran, et al. 2012), to the empirical validation section and showed that our algorithm performs favorably.

* We improved the clarity by fixing the typos highlighted by the reviewers and improving the notation.

---

### Decision · Program_Chairs · 2021-01-07
**Final Decision**

**Decision:**

Reject

**Comment:**

All reviewers feel this paper addresses and important topic, and has many merits. However, it is difficult to recommend publication at this time. The primary concern is that the paper has its theoretical optimality as an important contribution, but the reviewers and myself (in a non-public thread) were unable to verify the correctness of the proofs. In part unfortunately this is due to edits to the proofs happening late in the revision period, too late for further discussion with the authors. Some of the particular questions in the proof of theorem 1 (appendix B) include: clarifying the value of $\rho$ which makes the unnumbered equation above equation (6) equivalent to definition 1, and in particular whether the $1/|X_k|$ term should be inside or outside the absolute value; and clarifying various undefined symbols which are introduced in the equation at the top of page 13, but are never defined, including $M$, $b$, and $z_i$. Reviewers also had some concern that the algorithm should be benchmarked against more recent / better performant baselines than Kamiran et al. (2012).